# Wire Electrodes Embedded in Artificial Conduit for Long-term Monitoring of the Peripheral Nerve Signal

**DOI:** 10.3390/mi10030184

**Published:** 2019-03-13

**Authors:** Woohyun Jung, Sunyoung Jung, Ockchul Kim, HyungDal Park, Wonsuk Choi, Donghee Son, Seok Chung, Jinseok Kim

**Affiliations:** 1Center for Bionics, Biomedical Research Institute, Korea Institute of Science and Technology, Seoul 02792, Korea; jwooh1494@gmail.com (W.J.); T15333@kist.re.kr (S.J.); kimo@kist.re.kr (O.K.); hyungdal@kist.re.kr (H.P.); wonsuk@kist.re.kr (W.C.); 2School of Mechanical Engineering, Korea University, Seoul 02841, Korea

**Keywords:** neural interface, wire electrode, peripheral nerve electrode, artificial conduit, long-term implantation, neural signal recording

## Abstract

Massive efforts to develop neural interfaces have been made for controlling prosthetic limbs according to the will of the patient, with the ultimate goal being long-term implantation. One of the major struggles is that the electrode’s performance degrades over time due to scar formation. Herein, we have developed peripheral nerve electrodes with a cone-shaped flexible artificial conduit capable of protecting wire electrodes from scar formation. The wire electrodes, which are composed of biocompatible alloy materials, were embedded in the conduit where the inside was filled with collagen to allow the damaged nerves to regenerate into the conduit and interface with the wire electrodes. After implanting the wire electrodes into the sciatic nerve of a rat, we successfully recorded the peripheral neural signals while providing mechanical stimulation. Remarkably, we observed the external stimuli-induced nerve signals at 19 weeks after implantation. This is possibly due to axon regeneration inside our platform. To verify the tissue response of our electrodes to the sciatic nerve, we performed immunohistochemistry (IHC) and observed axon regeneration without scar tissue forming inside the conduit. Thus, our strategy has proven that our neural interface can play a significant role in the long-term monitoring of the peripheral nerve signal.

## 1. Introduction

Advances in interactive human–machine interfaces such as artificial prosthetic limbs and rehabilitation robots have been gradually attracting significant attention due to the new opportunities these present for enabling patients who have lost their arms or legs to improve their quality of life. Such approaches in human–machine interfaces are ordered in terms of the interface’s increasing degree of invasiveness: non-invasive, minimally invasive, and fully invasive applications [1]. As a representative example of non-invasive methods, electromyography (EMG) measurement has been widely utilized to control prosthetic limbs by monitoring the EMG signals of the muscle groups at the amputated region [2,3]. Although the simple attachment of EMG electrodes onto the amputee’s skin is non-invasive, significantly fewer distinct EMG signals can be obtained from the limited muscle groups at the amputation site compared to that of the peripheral nerves that carry neural signals originally intended to control precise motions ranging from finger movements to subtle tremors of the missing limb. Thus, many research groups have focused on fully invasive “sieve-type” electrodes to obtain numerous distinguishable signals by interfacing directly with the peripheral nerve [4,5]. As expected, different types of side effects have hindered the application of such extreme approaches to clinical surgery for long-term monitoring. The trade-off between the benefits of non-invasive and fully invasive methodologies should be optimized.

To overcome such critical issues, minimally invasive electrodes such as transverse intra-fascicular multichannel electrodes (TIME) [6] and longitudinal intra-fascicular electrodes (LIFE) [7,8] that have mechanical reliability and biocompatibility have been explosively developed for use in long-term implantation. However, there are severe problems such as scar formation caused by the inflammations that occur chronically around the implanted struts [9,10,11,12,13] (Figure 1A). In the case of the flexible penetrating microelectrode array (FPMA), which successfully acquires neural signals in the peripheral nerves, the signal-to-noise-ratio (SNR) was decreased due to the fibrous encapsulation around the electrode a week after its implantation [14]. As a promising candidate for solving the signal attenuation issues, neurotrophic electrodes (NE) have been suggested [15,16]. Wire-type neural electrodes were placed inside a glass conduit that was filled with a nerve growth factor (NGF). The NEs were successfully implanted and used to acquire neural signals from human brains for about four years. By filling it with the NGF, the neural signals were improved by narrowing the distance between the electrodes and the neurons by promoting nerve growth into the conduit over time. In addition, the internal and external nerve tissues of the conduit were connected together. Therefore, the neural electrodes inside the cone-like conduit permitted long-term implantation and high-quality signal acquisition by keeping the neural electrodes from glia scar formation. In addition, Lacour et al. developed a new type of stretchable neural device to overcome the limitations of chronically implanted electrodes that result in inflammatory responses and tissue scarring [17,18,19]. However, a few obstacles remain in practical applications. Manually fabricating the NE to have uniform electrical and mechanical performances is arduous and inefficient. Furthermore, the rigidity of the glass conduits used in the NEs is much stiffer than that of nerve tissue, which can lead to damage due to the mechanical mismatch between the tissue and the device. Therefore, a parylene sheath electrode (PSE)—fabricated using flexible polymers instead of rigid ceramics—was developed [20,21,22]. Although the PSE’s mechanical stiffness is reduced to compensate for the mechanical mismatch between the tissue and the PSE, it still has critical problems such as poor adhesion between the various evaporated metal films, which may cause an electrical breakdown due to its delamination from the substrate in long-term use. In addition, these NEs were only used in the rat motor cortex. Realizing the next generation of artificial prosthetic limbs will urgently require the application of flexible NEs to peripheral nerves.

In this case, we describe wire electrodes with thin and cone-shaped artificial conduits for long-term monitoring of the peripheral nerve. Our novel sensing platform can maintain reliable signal acquisitions by protecting the wire neural electrode from the typical formation of scar tissue that occurs a few weeks after implantation (Figure 1B). Au-plated Ni-Cr wires were placed inside the polyimide (PI)-based artificial conduit and the internal space was filled with collagen so that the axons of the nerve could regenerate through the conduit, which enables the chronic monitoring of neural activity. We implanted the fabricated wire electrodes with the artificial conduit into the sciatic nerve of rats with minimal invasiveness. After long-term neural signal acquisition, an IHC was carried out to confirm scar formation and nerve regeneration around the implanted site. Our wire electrode equipped with the cone-shaped artificial conduit is expected to pave the way for future human–machine interfaces in prostheses.

## 2. Materials and Methods

### 2.1. Overall Design of the Wire Electrodes with an Artificial Conduit

The artificial conduit was designed to protect the electrodes from scar formation due to nerve injury during implantation. The conduit was fabricated out of flexible and biocompatible PI. In addition, the Ni-Cr wires for recording neural signals were placed inside a cone-shaped PI conduit and connected to the copper (Cu) wire and then to a head stage. In total, three wire electrodes were used for signal acquisition.

### 2.2. Fabrication of Wire Electrodes with Artificial Conduit

Figure 2A shows the fabrication process for the artificial conduit. The structure of the cone-shaped artificial conduit was fabricated by applying heat and tensile force to the glass tube (Borosilicate Glass Capillaries, World Precision Instruments Inc., Sarasota, FL, USA) using a micropipette puller (P-97, SUTTER INSTRUMENT Co., Novato, CA, USA). Then, the dextran solution (20% w/w, SIGMA-ALDRICH, St. Louis, MI, USA) was thinly coated as a sacrificial layer on to the glass cone to later separate the PI (VTEC™PI-1388, RBI Inc., Toronto, ON, Canada) from the glass cone. The dextran-coated glass cone was cured at 65 °C for five minutes and the end of the glass cone was dipped into the precursor solution of PI for thin coating. This was then cured in an oven at 200 °C for two hours. After curing, a suture thread (8-0, NK825PDN, AILEE Co., Ltd., Busan, South Korea), which acts as an insertion guide during implantation, was fixed in the longitudinal direction using an epoxy (EPO-TEK®301, Epoxy Technology, Inc., Billerica, MA, USA). In this case, the two parts of the suture thread were tied in knots to increase the adhesion between the cone-shaped PI conduit and the thread, which increases the epoxy contact area. Epoxy was cured in an oven at 65 °C for two hours. After fixing the suture thread to the cone, it was immersed in deionized water (DI water) to dissolve the previously coated dextran layer and detach the conduit from the glass cone. The narrow and wide portions of the conduit that were fixed to the suture thread were trimmed to a total length of about 2 mm.

Where the metal electrode was exposed to nerve fibers, a 3-cm long Ni-Cr wire (KANTHAL precision Technology Inc., Hallstahammer, Sweden) of diameter 12.7 μm with aromatic PI (PAC 240, film insulation made of polyimide resins) was used as an insulation film. The insulation film at the tips of the wires of length ~200 μm was removed using oxygen plasma (Figure 2B). The exposed areas of the Ni-Cr wires were electro-plated with Au to improve their electrical properties and biocompatibility simultaneously [23,24,25]. The Ni-Cr wires were coiled into a spiral structure to cope with nerve and muscle movements. A Cu wire (SBYC-05, SME Co., Ltd., West Sussex, UK) of 36 AWG (american wire gauge) and the other end of the wire electrodes were connected by soldering. The head stage connector (Molex Co., Ltd., Lisle, IL, USA) was connected to the other end of the Cu interconnects by soldering for periodic neural signal measurement. To prevent the interconnect fracture and short-circuiting due to the penetration of bodily fluids into the wires during implantation, the Cu interconnection between the Ni-Cr electrodes and the head stage was encased in a silicone tube. Both ends of the silicon tube were sealed using epoxy and gingival mask (Esthetic Mask, DETAX GmbH&Co.KG, Ettlingen, Germany). Three wire electrodes were fixed inside the conduit in the longitudinal direction at 200 µm intervals with epoxy to prevent direct current flow between them (Figure 2C). To fix the silicon tube on the epineurium of the sciatic nerve, a PI film was laser-cut into a cruciform 3 mm in width and 3 mm in length. It was then pre-bent and fixed onto the silicon tube using a gingival mask (Appendix A). The wire neural electrode with the artificial conduit was filled with collagen. Since collagen forms into a gel within a few minutes at room temperature, the collagen injection was conducted on ice to keep the process as cold as possible. 

### 2.3. Electrical Characterizations

Electrochemical impedance spectroscopy (EIS) was used to analyze the electrical characteristics of the fabricated neural electrodes with conduit. The electrochemical impedance of the neural electrode was measured using a potentiostat (Versa STAT3, AMETEK Inc., Berwyn, IL, USA) over the frequency range of 1 Hz to 100 kHz in phosphate buffered saline (PBS) (1XPBS, SAMCHUN PURE CHEMICAL Co., Ltd., Seoul, South Korea). The amplitude was a 10 mV root mean square (RMS) for the potentiostatic EIS experiment. The electrochemical impedances were recorded with the two-electrode method using a Platinum (Pt) wire (RDE0021, ATfrontier Inc., Anyang, South Korea) as a reference electrode and the Au-plated Ni-Cr wire electrodes inside the conduit.

### 2.4. In vivo Implantation

The wire neural electrode with the artificial conduit was implanted into the sciatic nerve of rats (Figure 3). Eight-week-old Sprague Dawley (SD) male rats were anesthetized by an intramuscular injection of Zoletil (Virbac)-Rompun (BAYER) mixture at a 3:1 ratio. The sciatic nerve was exposed and the epineurium was incised longitudinally to make a 5 mm-long slit. The needle of the 8-0 surgical suture thread attached to the artificial conduit was inserted into the slit and pulled out through the non-incised nerve, which placed the artificial conduit inside the nerve. The artificial conduit was fixed by suturing and the slit on the nerve was sutured closed. To fix the artificial conduit and the silicon tube, arch-shaped PI film was sutured to the sciatic nerve by knotting the thread through the epineurium and the suture hole of PI film together. The ground electrode was placed near the muscle and the silicon tube was placed along the subcutaneous path to the head and exposed through the incision line at the back of the neck. The surgical site was sutured with a 3-0 suture thread. Before fixing the head stage, an incision was made on the scalp. All tissues around the skull were removed and the surface of the skull was sanded with the hand drill to attach the dental cement firmly. To increase the binding force, four screws were tightened into the skull at each of the four quadrants, which has their crossing point for the vertical and horizontal lines at the bregma (Appendix A). After the skull surface had completely dried, dental cement was applied between the skull and the head stage. The scalp was sutured closed around the head stage and the head stage was filled with the removable gingival mask to prevent contamination. Animal care and surgical procedures complied with the Institutional Animal Care and Use Committee guidelines.

### 2.5. Measurement Set-up

A SmartBox (NeuroNexus Inc., Ann Arbor, MI, USA) was used to record neural signals. The SmartBox has a 64-channel connector (Smart Link 64ch, NeuroNexus Inc.) with two 32-channel male Omnetic connectors for access. Therefore, an interconnection cable was fabricated to go between the head stage of the rat and the Omnetic connectors. One side of the cable consists of a 32-channel omnetic connector (female) and the other side consists of a five-channel molex connector for connecting to the head stage of the rat. All recorded signals were sampled at 20 kHz and through a band-pass filter with the range set to 30 Hz–5 kHz. After setting the equipment for recording neural signals, the rat was anesthetized using isoflurane. The neural signal measurement equipment and head stage fixed to the skull were connected for signal acquisition by using the interconnection cable. Attempts were made to evoke sensory single unit action potential via mechanical stimulation (Figure 4B). The sole of the electrode-implanted leg was mechanically stimulated using a cotton swab and brush.

### 2.6. Immunohistochemistry

For IHC, the neural electrode-implanted SD rats were perfused with 4% paraformaldehyde (PFA) in PBS. Then, the sciatic nerves were extracted and fixed in 4% PFA for three days at 4 °C. Tissues were then dehydrated with sucrose solution at 4 °C. Nerves were frozen in an optimal cutting temperature compound (OCT compound) (FSC22, Leica Biosystems Richmond, Inc., Dublin, Island) at −80 °C overnight and then frozen-sectioned longitudinally at 10 µm-thick intervals. The sections were attached to saline-coated micro slides (5116-20F, MUTO Pure Chemicals Co., Ltd., Tokyo, Japan) and dried for a few minutes. The OCT compound was washed in PBS three times and the sections were fixed with 4% PFA solution for 20 minutes at 4 °C and washed again with PBS three times. The sections were permeabilized in 0.2% Triton X-100 for 15 minutes, washed three times in PBS, and blocked overnight in 4% Bovine serum albumin (BSA) at 4 °C. After blocking, the sections were incubated overnight with the following primary antibodies: mouse anti-neurofilament medium (ab7794, Abcam Inc., Cambridge, UK) and rabbit anti-fibronectin (ab2413, Abcam Inc.). After washing in 1% BSA, the samples were incubated for two hours at room temperature with the appropriate secondary antibodies: AlexaFluor 594-donkey anti-rabbit IgG (ab150076, Abcam Inc.) and AlexaFluor 488 anti-mouse IgG (ab150113, Abcam Inc.). Preparations were washed in 1% BSA three times and incubated for 10 minutes at room temperature with 4′,6-diamidino-2-phenylindole (DAPI) (Thermo Fisher Scientific Inc., Waltham, MA, USA) in the dark to stain the cell nucleus. Sections were washed in 1% BSA and PBS once each and preserved with mountant (ThemoFisher Scientific Ltd.). Images were visualized using a confocal microscope (Leica Microsystems, Wetzler, Germany).

## 3. Results and Discussion

### 3.1. Structural Design and Intrinsic Properties of the Wire Electrodes with the Artificial Conduit

To protect the Au-plated wire electrodes from scar formation that usually forms around the electrodes after implantation, we applied the cone-shaped artificial conduit to the wire electrodes. The artificial conduit was fabricated using PI that is both flexible and biocompatible. The diameter of the narrow portion of the artificial conduit was ~200 μm and the wide portion was ~500 μm. The artificial conduit that adhered to a suture thread was fully inserted into the sciatic nerve during implantation and the aperture of the conduit played a role in allowing the respective, disconnected axons to be connected. The thickness (~15 μm) of the artificial conduits was controlled by dipping in liquid PI. The cone-shaped structure of the PI artificial conduit was mechanically reliable for maintaining its form factor when compressively strained due to its high elastic modulus (~2.5 GPa). The neural wire electrode with the artificial conduit consisted of a head stage that could be connected to the external neural signal measurement equipment for real-time signal measurements (Figure 5). The electrical characteristics of the fabricated wire neural electrodes with the artificial conduit were evaluated using EIS. According to the measurement results, the impedances before and after electroplating with Au were 156.2 kΩ (Ni-Cr) and 38.5 kΩ (Au-Ni-Cr) at 1 kHz, respectively (Figure 4A). Previous reports support that the lowered impedance of our Au-Ni-Cr wires under 1 MΩ was suitable for recording neural signals [25,26]. We confirmed that the electro-plated Au allowed the peripheral nerves to avoid Ni-Cr alloy-induced nickel allergy and toxicity problems (Figure 6) [23,24,27]

### 3.2. In vivo Preparation and Long-Term Measurement for Peripheral Neural Signals

Each neural electrode was composed of three wires and an artificial conduit. Each of the five SD rats had an electrode implanted into their sciatic nerve (Figure 5). Using a suture thread fixed to the conduit, the neural electrodes were inserted into and fixed to the sciatic nerves (Figure 3B). A headstage was fixed to the skull for periodic signal measurement. The adhesion of dental cement alone could not hold the head stage in the long term. Therefore, the head stage was fixed more firmly by applying skull screws to the skull in advance. The interconnection cables bridging between the sciatic nerve and the headstage were placed subcutaneously inside a silicone tube to be protected. After implanting the wire neural electrode, we used a neck collar to prevent self-injury such as toe-biting (Appendix A).

After implanting the five neural electrodes, we could not immediately obtain neural signals (Appendix A). This was expected since the damaged axons had not yet regenerated into the artificial conduit and established connections between the proximal and distal axons. However, a low noise level was identified (about 50 μV) that was suitable for acquiring neural signals. After 19 weeks, we obtained neural signals from only two rats (Figure 4C and Appendix A). The single waveforms indicating sensory neural signals in each graph were almost identical due to the possibility that the same nerve fibers were making contact with or near two or three wires simultaneously (Figure 4C and Appendix A). The neural signals were well synchronized with mechanical stimulations given to the foot using a cotton swab and brush. This result is comparable to those of previous reports (Figure 4 and Figure 5, Appendix A) [28]. Unfortunately, we could not monitor neural signals from the other three rats due to undesired incidents such as damage to the Cu interconnection cables or headstage modules (Appendix A). The Cu interconnection cables inside the silicon tube were broken due to mechanical fatigue that originated from the continuous movement of the rat’s neck. This issue could be significantly improved by making the Cu interconnection cables more flexible and stretchable. In addition, when we periodically measured neural signals, the delamination of the head stage from the rat’s head could be due to its low adhesion. Therefore, the interconnection cable and the head stage are very important components for long-term signal measurement. If these issues are optimized, our chronic neural platform would be significantly improved.

### 3.3. Immunohistochemistry

The wire neural electrodes with artificial conduit inserted in the sciatic nerve were double stained with anti-neuro-filament medium (NFM) antibody and anti-fibronectin antibody to demonstrate the regenerating axons into the PI cone-shaped conduit and observe the formation of neuronal scarring (Figure 6). The sample was also stained with DAPI for nuclear acid to see the cells’ general location. Six months after surgery, it was found that newly sprouted axons had extended and filled the space of the conduit. Recently grown axons showed different extending tendencies from pre-existing neuronal axons, which formed a cone shape. Meanwhile, fibronectin, which is the structural molecule of neuronal scar tissue, was found just outside the conduit. More importantly, scar tissue scarcely existed inside the conduit cavity. Fibronectin was highly expressed at the upper side of the conduit due to the surgical procedure. As mentioned, to insert the conduit, a longitudinal neural incision was made to form a gap. In this procedure, neuronal injury was made to cause neuronal scarring in the long-term. Nevertheless, it is shown that this neural scarring from the surgical procedure does not invade the inside of the conduit. There is visible fibronectin at the lower side of the cone because fibronectin is also a major component of the peripheral nerve epineurium [29]. Therefore, it can be concluded that the wire electrode placed inside the conduit was protected from scar tissue, which further verifies the chronic neural signal acquisition feasibility of the neural electrode with the artificial conduit. This section may be divided into subheadings.

## 4. Conclusions

For the ultimate purpose of the long-term implantation of neural electrodes, a peripheral nerve electrode with a cone-shaped artificial conduit protecting the wire electrodes from scar formation was developed. The conduit was fabricated using a flexible and biocompatible material, PI, and Au-plated Ni-Cr wires. The neural electrode was implanted directly into the sciatic nerve of the rat, and, after 19 weeks, neural signals were obtained during mechanical stimulation to the sole of the rat. IHC analysis showed that scar tissues were formed outside the conduit, but no scars were found inside the conduit. Nerve fibers grew into the conduit in both directions and connected to each other to make contact with the wire electrodes. Therefore, neural signals could be acquired. Using the cone-shaped conduit structure, chronic signal acquisition could be achieved by preventing performance degradation in the electrode due to scar-tissue formation. Furthermore, we have attempted to find a way to further accelerate nerve regeneration through experiments with various hydrogels filling the artificial conduit. We expect that this will be applied to the development of prosthetic hands with movements similar to that of natural hands by overcoming the limitation on the low number of distinguishable movement intentions in existing EMG prosthetic arms and solving the neural signal attenuation problem in the long-term implantation of neural interfaces.

## Figures and Tables

**Figure 1 micromachines-10-00184-f001:**
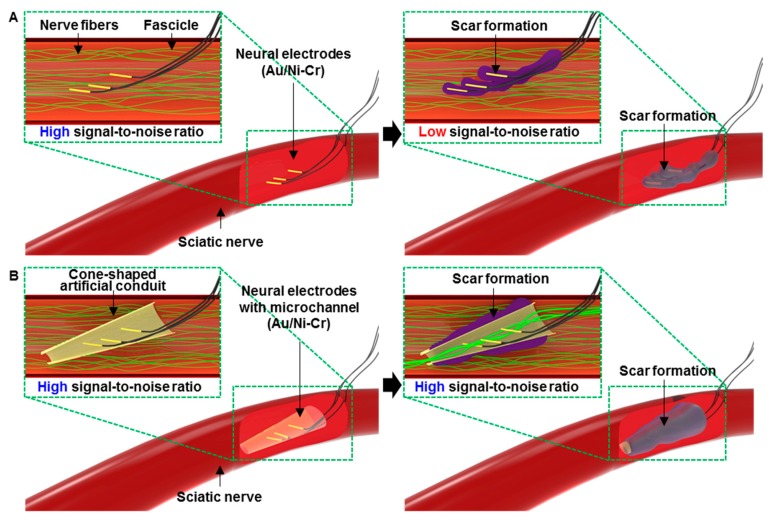
Overall schematics and *in vivo* images of wire electrodes with the artificial conduit. (**A**) Schematic of implanted wire electrode without the artificial conduit into the sciatic nerve. (**B**) Schematic of implanted neural electrode with the artificial conduit into the sciatic nerve.

**Figure 2 micromachines-10-00184-f002:**
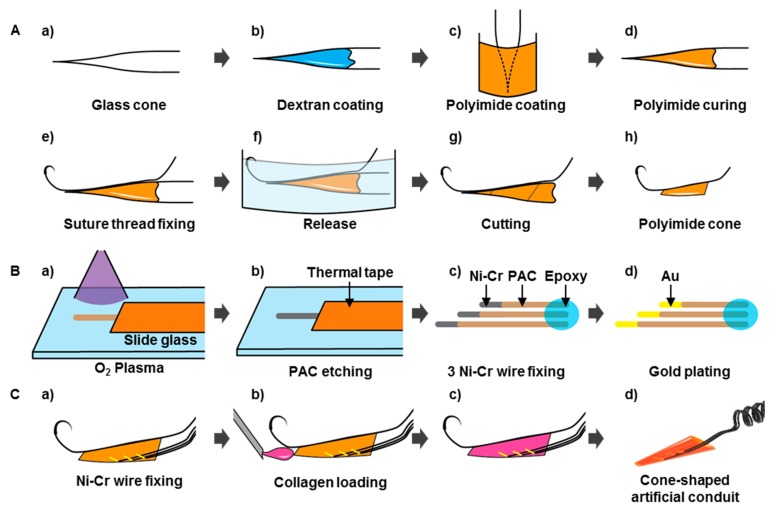
Fabrication and implantation process for wire electrodes with artificial conduits. (**A-a**) Preparation of glass cone. (**A-b**) Dextran coating. (**A-c,d**) PI coating and curing. (**A-e**) suture thread fixation. (**A-f**) PI cone release. (**A-g,h**) PI cone cutting. (**B-a,b**) Ni-Cr wire insulation removal by plasma etching. (**B-c**) Three Ni-Cr wires array and fixation using epoxy. (**B-d**) Au electroplating. (**C-a**) Attaching Ni-Cr wires within the PI cone. (**C-b,c**) Collagen loading. (**C-d**) Fabricated neural electrode with the cone-shaped artificial conduit.

**Figure 3 micromachines-10-00184-f003:**
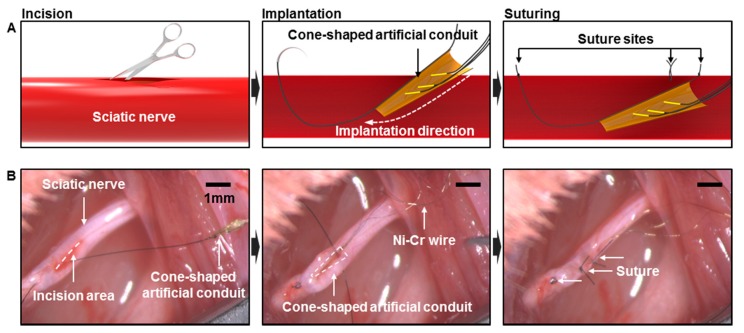
Implantation process of wire electrodes with the artificial conduit. (**A**) Schematics of the implantation process. (**B**) *in vivo* images of the implantation process. Longitudinal incision of the sciatic nerve and implantation of the conduit using the suture thread, fixation of the conduit by suturing.

**Figure 4 micromachines-10-00184-f004:**
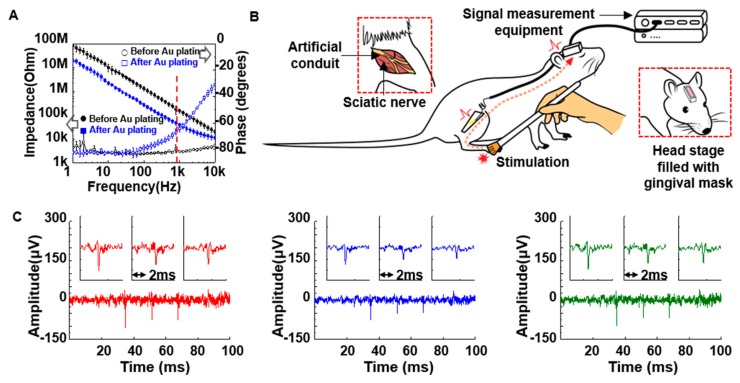
Neural signal measurement. (**A**) Electrochemical impedance and phase changes across the frequency of the neural electrode with the artificial conduit. The impedances before and after electro-plating were 156.2 kΩ and 38.5 kΩ at 1 kHz. (**B**) Schematic for the neural signal recording setup. (**C**) Results of neural signal acquisition for the three wire electrodes.

**Figure 5 micromachines-10-00184-f005:**
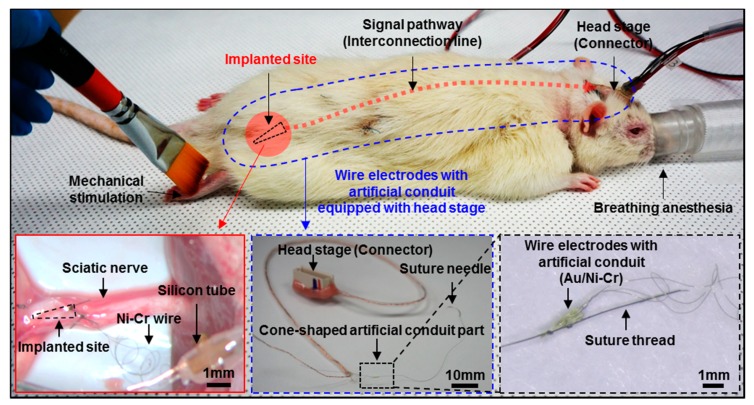
Overall *in vivo* images of wire electrodes with the artificial conduit. Implantation and neural signal measurement of fabricated neural electrodes with the artificial conduit.

**Figure 6 micromachines-10-00184-f006:**
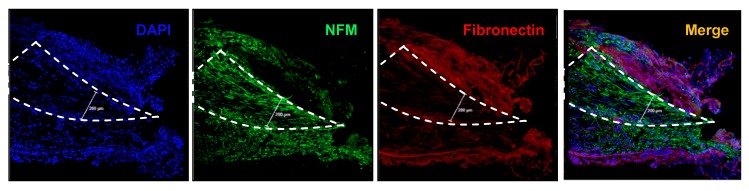
*IHC analysis*. Immunohistochemistry images of the representative longitudinal section of the sciatic nerve implanted with the neural electrode and artificial conduit.

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
