# Peer review of "Wire Electrodes Embedded in Artificial Conduit for Long-term Monitoring of the Peripheral Nerve Signal"

_micromachines, 2019, doi:10.3390/mi10030184_

Round 1

Reviewer 1 Report

In this paper the authors chronically implanted Au plated Ni-Cr wires in the rat sciatic nerve and performed long-term recordings. The wires were encased inside a Polyimide-based (PI) artificial conduit, which had a conical shape. It is argued that this inhibits scar formation close to the wires and instead localizes the fibrotic tissue on the outside of the PI conduit. The authors conducted electrochemical impedance spectroscopy to characterize the electrical response of the fabricated wires, before and after electroplating. The impedance values measured are expected for these types of electrodes. The authors also provide images of immunohistochemical staining with antibodies against neurofilament (NFM) and fibronectin.

The methodology is sound and the results are interesting. The Introduction is compact and even though the authors reference papers from different technologies (TIME, LIFE, FPMA, etc), they do not reference any work on stretchable electrodes, which reduce inflammatory responses and tissue scarring when implanted chronically. The authors should include and expand in their introduction, on a few prominent examples from the Lacour group (see below).

1)      Daniel J Chew, Lan Zhu, Evangelos Delivopoulos, Ivan R Minev, Katherine M Musick, Charles A Mosse, Michael Craggs, Nicholas Donaldson, Stéphanie P Lacour, Stephen B McMahon, James W Fawcett, “A microchannel neuroprosthesis for bladder control after spinal cord injury in rat”, Science translational medicine 5, 2013.

2)      Evangelos Delivopoulos, Daniel J Chew, Ivan R Minev, James W Fawcett, Stéphanie P Lacour, “Concurrent recordings of bladder afferents from multiple nerves using a microfabricated PDMS microchannel electrode array”, Lab on a chip 12, 2012.

3)      IR Minev, DJ Chew, E Delivopoulos, JW Fawcett, SP Lacour, “High sensitivity recording of afferent nerve activity using ultra-compliant microchannel electrodes: an acute in vivo validation”, Journal of neural engineering 9, 2012.

Furthermore, the authors should address the following points in their manuscript, before acceptance:

a)      There are multiple minor linguistic errors throughout the manuscript (e.g. line 48 “...biocompatibility have been explosively” developed…, line 209 “mountant” etc”). The manuscript should be edited and all minor errors corrected. Also in line 279 there is a phrase, probably from a draft version of the document that should be removed (“This section may be divided by subheadings”).

b)      The authors mention that the wires are Ni-Cr based. Chromium is toxic and problematic in long term implantations. This point should be addressed, preferably in the “Results and Discussion” section. In relation to this, the authors do mention on line 229 that the Au plating allows peripheral nerves to avoid the Ni-Cr interface. However, this prevents only direct contact, as long as the Au layer is intact. It does not prevent leakage of toxic Cr molecules into the ECM, nor is it a viable long term solution. A statement explaining this should be made.

c)       Maybe I missed it, but I could not find the exact dimensions of the conical PI conduit: diameter of base and vertex and height. These dimensions should be included. Also, a small diagram in Figure 2, with the dimensions of the electrodes/wires would be extremely helpful. This information is probably lost in the text.

d)      In Figure 5 panels A and B are the other way around. Also, the authors include 3 recordings with 3 different colours: red, blue and green. I did not understand if these were recordings from different animals or different electrodes/devices. All this is further confounded, when the authors mention in line 248 that signals were recorded only from 2 rats.

e)      In Figure 5 the authors provide the impedance modulus, but not the phase of the electrodes they tested. It would be good to see how the phase changes across the frequency, to understand whether there is a capacitive effect with these wires and how it is attenuated. Also, is this the average spectra from multiple devices, or one?

f)       I could not access the supplementary material, so I cannot comment on what the authors have submitted there.

As a suggestion, the authors should also examine collagen and vimentin as markers for fibrosis. Even though fibronectin is present during scar tissue formation, it is not the only molecule implicated in the process. There are also myofibroblasts (depending on the tissue) and collagen present. Staining with anti-vimentin and anti-collagen type I/II antibodies would strengthen the authors’ narrative on absence of scar tissue within the conical conduit. I realize however, this is time consuming and difficult within the context of this study.

Author Response

Response to Reviewer 1 Comments

Point 1: The methodology is sound and the results are interesting. The Introduction is compact and even though the authors reference papers from different technologies (TIME, LIFE, FPMA, etc), they do not reference any work on stretchable electrodes, which reduce inflammatory responses and tissue scarring when implanted chronically. The authors should include and expand in their introduction, on a few prominent examples from the Lacour group (see below).

Response 1: Thank you for your comment. Three references were added to the introduction part of this manuscript and highlighted in red.

Point 2: The authors mention that the wires are Ni-Cr based. Chromium is toxic and problematic in long term implantations. This point should be addressed, preferably in the “Results and Discussion” section. In relation to this, the authors do mention on line 229 that the Au plating allows peripheral nerves to avoid the Ni-Cr interface. However, this prevents only direct contact, as long as the Au layer is intact. It does not prevent leakage of toxic Cr molecules into the ECM, nor is it a viable long term solution. A statement explaining this should be made.

Response 2: According to S. Choi et al., they covered the Ag wire with Au shell to prevent the inflammation and cytotoxicity due to silver diffusion. When they estimated the diffusion rate of the silver particle, there was almost no diffusion. Also, in our study, issues related to biocompatibility due to the the use of Cr wire electroplated with Au was not observed. We added the reference to the manuscript and highlighted in red.

Point 3: Maybe I missed it, but I could not find the exact dimensions of the conical PI conduit: diameter of base and vertex and height. These dimensions should be included. Also, a small diagram in Figure 2, with the dimensions of the electrodes/wires would be extremely helpful. This information is probably lost in the text.

Response 3: As described in the “Results and Discussion” section, the exact dimensions of the conical PI conduit is as follows. A diameter of the base (the wide portion of the cone): ~500 , vertex (the narrow part of the cone): ~200 , and height (total length): 2 mm.

Point 4: In Figure 5 panels A and B are the other way around. Also, the authors include 3 recordings with 3 different colours: red, blue and green. I did not understand if these were recordings from different animals or different electrodes/devices. All this is further confounded, when the authors mention in line 248 that signals were recorded only from 2 rats.

Response 4: Panels A and B are modified in the right way and highlighted in red. Also, the result of recording described in figure 5 is from one device. More recording data from 2 rats are described in the supplements.

Point 5-1: In Figure 5 the authors provide the impedance modulus, but not the phase of the electrodes they tested. It would be good to see how the phase changes across the frequency, to understand whether there is a capacitive effect with these wires and how it is attenuated.

Response 5-1: Thank you for your comment. Graph of phase changes of electrodes across the frequency was added to figure 5A, and changed captions were highlighted in red.

Point 5-2: Also, is this the average spectra from multiple devices, or one?

Response 5-2: Average spectra is from 3 electrode wires in one PI conduit device.

Point 6: I could not access the supplementary material, so I cannot comment on what the authors have submitted there.

Response 6: I added the supplementary data from the next page of this document.

Reviewer 2 Report

This study describes the synthesis and function of a flexible cone with embedded recording electrodes used in the peripheral nervous system of rats. The hypothesis was that the cone 'shield' would prevent scar formation from interfering with nerve recordings at the electrodes.

Overall the manuscript describes some progress toward this goal. The method for electrode/collar synthesis is highly technical and shows impressive technical skill. But the end product seems to be less robust than would be necessary for long term use in a practical context. The failure of the copper connections and the inability to retain the headstage in position are significant hurdles.  In addition not all of the electrodes appeared to function (that is, not all within the set of 3).

Whilst an admirable set of experiments my concern is that there is no statistical analysis of the results.  Is this meant to be a pilot study? It reds like that to me. As an early proof of concept study this is promising but further analysis using statistical methods should be done to clarify the outcome.

Although the authors show evidence of neural signals through the implants they should report how many of the electrodes were functioning at the various time points studied. They state that they did not detect signals early on but at later stages they did.  The time course of this should be mapped onto time course for nerve 'regeneration' into the cone.

The authors used an antibody to fibronectin to look at the scar, which is interesting, but I wonder why they did not use an antibody to collagen 1 (the matrix loaded into the cone) to see how long it was retained (or whether it was retained) at various time points. 

The authors claim that nerves regenerated through the cones based on a combination of nerve conduction signalling detected by their implants and also by neurofilament labelling. I feel this is not sufficiently robust to claim regeneration.  I think a time lapse sequence, perhaps using a neuro-tracer dye to show that the axons extend through the cone to reconnect would be needed.

On another note, I am not clear how many electrodes were implanted in each animal (sets of three). It shows one in Figure 4 and Figure 5 but on P 8 L235 it states that 5 neural electrodes were placed in each sciatic nerve. This needs to be clarified.

I applaud the author's technical achievement but the interpretation and presentation of the data should be sharpened up in preparation for publication.

The authors should expand more on how the limitations of the method could be improved for their expected use in human prosthetic limbs.

Author Response

Response to Reviewer 2 Comments

Point 1: Although the authors show evidence of neural signals through the implants they should report how many of the electrodes were functioning at the various time points studied. They state that they did not detect signals early on but at later stages they did.  The time course of this should be mapped onto time course for nerve 'regeneration' into the cone.

Response 1: Thank you for your comment. 5 PI conduits are transplanted to the right sciatic nerve of each five rats. Unfortunately, due to the undesired damage caused to the headstage or Cu interconnection cables, only 2 of total conduits were able to record the nerve signal. Also, there were some difficulties with regular and frequent signal recording experiment because animals had to be completely anesthetized. There was a lack of our resources, but we will record the signal from neural electrodes in early stages and map then onto time course for a demonstration of nerve regeneration in our further work.

Point 2: The authors used an antibody to fibronectin to look at the scar, which is interesting, but I wonder why they did not use an antibody to collagen 1 (the matrix loaded into the cone) to see how long it was retained (or whether it was retained) at various time points.

Response 2: The role of loaded collagen1 inside the PI cone was to induce the axon regeneration inside the conduit. The more important thing was the prevention of scar formation of newly regenerated axons than retaining of collagen1. Therefore, confirming that there is no scar tissue formation inside the conduit was more focused to observe.

Point 3: The authors claim that nerves regenerated through the cones based on a combination of nerve conduction signaling detected by their implants and also by neurofilament labeling. I feel this is not sufficiently robust to claim regeneration.  I think a time lapse sequence, perhaps using a neuro-tracer dye to show that the axons extend through the cone to reconnect would be needed.

Response 3: Thank you for your opinion. Demonstration of reconnection of extended axons through the cone will be observed with neuro-tracer dye in our further work.

Point 4: On another note, I am not clear how many electrodes were implanted in each animal (sets of three). It shows one in Figure 4 and Figure 5 but on P 8 L235 it states that 5 neural electrodes were placed in each sciatic nerve. This needs to be clarified.

Response 4: 5 neural conduits were placed in each right sciatic nerve of 5 SD rats. Therefore, one conduit is transplanted to the right sciatic nerve of each SD rat. One conduit consists of 3 wire electrodes.

Point 5: I applaud the author's technical achievement but the interpretation and presentation of the data should be sharpened up in preparation for publication.

Response 5: We appreciate your comment. We will reflect reviewers’ comment in our futher work.

Point 6: The authors should expand more on how the limitations of the method could be improved for their expected use in human prosthetic limbs.

Response 6: Thank you for your comment. For expected use of PI conduit in human prosthetic limbs, we are planning our future works to develop advanced neural electrodes improved with followings: more biocompatibility, a neural interface that is similar to nerve modulus, and enhanced nerve regeneration with nerve regeneration factors.
